# Outcomes of Hospitalized Patients with Glucocorticoid-Induced Hyperglycemia—A Retrospective Analysis

**DOI:** 10.3390/jcm9124079

**Published:** 2020-12-17

**Authors:** Neele Delfs, Tristan Struja, Sandra Gafner, Thaddaeus Muri, Ciril Baechli, Philipp Schuetz, Beat Mueller, Claudine Angela Blum

**Affiliations:** 1Departments of General Internal and Emergency Medicine, Medical University Clinic, Kantonsspital Aarau, Tellstrasse 25, Haus 7, 5001 Aarau, Switzerland; dneele@hotmail.com (N.D.); tristan.struja@ksa.ch (T.S.); sandra.gafner@stud.unibas.ch (S.G.); thadde.muri@stud.unibas.ch (T.M.); happy.mueller@unibas.ch (B.M.); 2Department of Endocrinology and Diabetology, Medical University Clinic, Kantonsspital Aarau, Tellstrasse 25, Haus 7, 5001 Aarau, Switzerland; ciril.baechli@gmail.com (C.B.); philipp.schuetz@ksa.ch (P.S.)

**Keywords:** glucocorticoid-induced hyperglycemia, glucocorticoid-induced diabetes, outcome, cardiovascular events, mortality, infection, hypoglycemia

## Abstract

Background: Glucocorticoid (GC)-induced hyperglycemia is a frequent side effect in hospitalized patients. Guidelines recommend treat-to-target treatment between 6–10 mmol/L (108–180 mg/dL) with insulin, but data on outcome is scarce. We investigated the 30-day outcome in hospitalized patients receiving GCs. Methods: All patient records of hospitalized patients between January 2014 and April 2018 were screened for GC administration and consecutive hyperglycemia. The primary combined endpoint consisted of death, cardiovascular events, and infections until 30 days after admission. Hypoglycemia was a secondary outcome. Results: Of the 2424 hospitalized patients (9.6% of all hospitalized patients) who received systemic GCs and met inclusion criteria, the overall incidence for GC-induced hyperglycemia was 812 (33.5%), and 89 (3.7%) had at least one documented hypoglycemia during their hospital stay. Compared to patients with normoglycemia, GC-induced hyperglycemia had an adjusted-odds ratio of 1.68 (95% CI 1.25–2.26) for the combined primary endpoint. Hypoglycemia even had an odds ratio of 1.95 (95% CI 1.2–3.17). Conclusions: Mortality, cardiovascular events, and rate of infections were markedly higher in patients with GC-induced hyperglycemia as compared to patients with normoglycemia. Importantly, hypoglycemia was associated with a doubled risk for adverse outcome. Future studies should evaluate whether optimized glucose control by minimizing the risk for hypoglycemia has a beneficial effect on clinical outcomes in patients with GC-induced hyperglycemia.

## 1. Introduction

Glucocorticoids (GCs) are frequently used as anti-inflammatory agents [1], and their use has recently massively increased in patients with COVID-19 [2]. About 30% of patients treated with GCs will develop GC–induced hyperglycemia [3,4,5].

Until recently, guidelines recommended treat-to-target insulin with narrow target glucose levels for all hospitalized patients [6], due to the allegedly marked effect of the intensive care study by van den Berghe et al. [7]. Subsequent studies contrasted these findings in the presence of frequent, insulin -induced hypoglycemias [8]. The benefit of tight glucose control is even less clear in patients outside the ICU. Benefits of tight glucose control have only been shown for postoperative infection rate in surgical patients [9,10]. Target levels for general in-hospital hyperglycemia have been gradually increased to 7.8–10.0 mmol/L (140–180 mg/dL) until 2018 [8,11,12] due to the risk of hypoglycemia, which has been associated with increased cardiovascular events and mortality [6,13,14,15,16].

There are few studies which specifically focus on treatment of GC induced hyperglycemia, and most prospective studies investigate glucose control with different insulin regimens [5,12,17,18]. There is no specific data on patient-related outcomes such as mortality or cardiovascular events in patients with GC-induced hyperglycemia. These patients frequently receive only short-term GCs and therefore experience especially short-term hyperglycemia. Furthermore, the effect of treat-to-target insulin on outcome in GC-induced hyperglycemia has not been shown.

We aimed to assess the 30-day outcome of hospitalized patients with GC-induced hyperglycemia. We therefore investigated the incidence of adverse events (mortality, cardiovascular events, and infections) in patients with GC-induced hyperglycemia in this retrospective study.

## 2. Materials and Methods

This was a retrospective observational study. The local ethics committee (EKNZ, Ethics Committee of Northwestern and Central Switzerland) approved the study (EKNZ 2018-01271). The study adheres to the principles of the Declaration of Helsinki and to the STROBE statement [19].

Using the electronic medical charts and data of our clinical information system, we analyzed data of hospitalized patients between 1 January 2014 and April 2018 at the Internal Medicine Ward of a tertiary care hospital in Switzerland (Medical University Clinic, Kantonsspital Aarau, Aarau, Switzerland), who were treated with GCs. Inclusion criteria were age of at least 18 years, hospitalization for at least 3 days, and administration of at least 10 mg prednisolone equivalent per day. Exclusion criteria were multiple hospitalizations if the consecutive hospitalization was within 30 days of the index hospitalization. We assessed GC dose, indication for GC administration, glucose measurements, insulin dosing, pre-existing diabetes diagnosis and treatment, rate of death, cardiovascular events, infections within 30 days of admission, and rate of hypoglycemia while hospitalized. The 30-day interviews were available from an ongoing quality assessment [20,21].

### 2.1. Definitions and Categorizations

GC–induced hyperglycemia was defined as either morning fasting blood glucose of >7.0 mmol/L (126 mg/dL) or a random glucose measurement of >11.0 mmol/L (198 mg/dL) after the start of GCs, regardless of pre-existing diabetes status. Normoglycemia was defined as fasting blood glucose of 7.0 mmol/L (126 mg/dL) or lower, and random glucose of 11.0 mmol/L (198 mg/dL) or lower [6,12,22]. GC-induced diabetes was defined if the above criteria for hyperglycemia were met with no pre-existing diabetes.

The following events were counted as a cardiovascular event: cardio-ischemic event (acute coronary syndrome), acute decompensated heart failure needing hospital admission, and ischemic cerebrovascular event (acute stroke and transitory ischemic attack). Day 1 was defined as the first day of GC administration.

Glycemic variability was calculated as the coefficient of variation (CV) of glucose as follows:CV = SD/mean ∗ 100%.

Mean glucose differences from peaks to nadirs (MAGE) were calculated as well [23].

The percentage of glucose readings in the range was defined as the number of finger-stick glucose measurements between 4.0 and 10.0 mmol/L (72–180 mg/dL).

Hypoglycemia was defined as any blood glucose < 4.0 mmol/L (<72 mg/dL). GC dose was assessed as milligram of prednisolone equivalent per kilogram of body weight per day.

Four groups for GC administration groups were formed.

The Autoimmune/Inflammation group included the following diagnoses: neuroinflammatory disease, rheumatologic autoimmune disease, gout, vasculitis, allergies, dermatologic indications, chronic inflammatory bowel diseases, kidney transplants, and glomerulonephritis.

The hemato-oncology group included chemotherapy and antiemetic indications.

The infection/pneumology group included the following diagnoses: chronic obstructive pulmonary disease (COPD), airway infections, sepsis, and ear, nose, and throat infections.

The endocrinology group included 92 patients (79%) with adrenal insufficiency or stress prophylaxis, the remaining cases had hypercalcemia or amiodarone-induced toxicosis.

### 2.2. Endpoints

The objective of this study was to evaluate the outcome in hospitalized patients with GC-induced hyperglycemia. The primary endpoint was the combined endpoint of 30-day mortality, cardiovascular events, and infections. Secondary endpoints were each of the single items of the composite endpoint: 30-day mortality, 30-day cardiovascular events, 30-day infections, and in addition, in-hospital hypoglycemia.

### 2.3. Hypothesis

We hypothesized that the risk of mortality, cardiovascular events, and infections would be higher in the group with GC-induced hyperglycemia. We expected a higher rate of hypoglycemia in the group of patients with GC-induced hyperglycemia due to the therapeutic use of insulin. A systematic review and meta-analysis on glycemic control in non-critically ill patients by Murad et al. [9] found pooled relative risk ratios for death, infection, myocardial infarction, and stroke of 0.85 (95% CI 0.58–1.26), 0.41 (95% CI 0.21–0.77), 0.69 (95% CI 0.37–1.28), and 0.63 (95% CI 0.29–1.38), respectively. Based on their findings, we performed a Chi-squared two-proportions power calculation assuming an alpha level of 5%, an intervention group size of 812 with a rate ratio of 0.02, and a control group size of 1612 with a rate ratio of 0.039 (numbers based on Murad et al.), giving a power of 0.73.

### 2.4. Statistical Analysis

We did not use a test for normality such as the Shapiro–Wilk test, due to their tendency to give false-positive results in large samples. Instead, we checked graphically by plotting histograms and quantile–quantile plots. Discrete variables are expressed as counts (percentages) and continuous variables as either means (SD) if normally distributed or medians (IQR) if non-normally distributed. Continuous variables were analyzed using the Wilcoxon rank-sum test. Categorical and binary variables were analyzed using Fisher’s exact test.

We have only performed analysis with complete cases. Thus, we did not perform any imputation method for missing covariates, and cases with missing covariates were excluded from the multivariable analysis. Over the course of the hospitalization, a mean number of three blood glucose measurements per day was required.

Endpoints were evaluated by logistic regression modeling. First, three unadjusted models were fitted:(A)between patients with GC-induced hyperglycemia and normoglycemia(B)between patients with GC-induced diabetes and patients without diabetes who had normoglycemia(C)between patients with pre-existing diabetes and patients with GC-induced diabetes

Then, these three models were adjusted for age and pre-existing comorbidities by the age-adjusted Charlson Comorbidity Index, [24] by GC dose (mg prednisolone equivalent per kg body weight per GC day), reasons for GCs, mean glucose, CV of glucose, percent of glucose values in range, and hypoglycemia < 4.0 mmol/L (<72 mg/dL). The choice of adjusted variables was based on expert opinion and a review of the literature. The variables chosen were identified by previous research as important confounders [25,26,27].

An alpha level of 5% was deemed significant. Data was analyzed using Stata statistical software (Stata/MP 15.1, Stata Corp., College Station, TX, USA).

The hypoglycemic events were too rare for further evaluation in a regression model, but we accounted for this secondary endpoint by adding it as a cofactor in the model for the other endpoints.

## 3. Results

Of the 25,183 patients hospitalized at the Medical University Clinic from 1 January 2014 to 26 April 2018, 4060 patients (16%) received GCs at a dose of at least 10 mg prednisolone equivalent per day and were hospitalized for a minimum of three days. Out of these, in 2424 patients (9.6%), data was available for evaluation (see Figure 1 for study flow chart).

Of the 2424 patients, 1123 (46%) patients were men, the mean age was 67 years (SD ± 15), and pre-existing diabetes was present in 511 patients (21%; see Table 1 for baseline characteristics and Table A1 and Table A2 for detailed baseline characteristics per subgroup).

Eight hundred and twelve patients (33.5%) had GC-induced hyperglycemia, of which 399 had GC-induced diabetes, and 413 had a pre-existing diabetes. 1612 patients remained normoglycemic, of which 98 had a pre-existing diabetes. The rate of hypoglycemia was 9.6% (*n* = 78) in patients with GC-induced hyperglycemia6 (see also Table 2). Mean GC dose (prednisolone equivalents/kg/day) was higher in patients with GC-induced hyperglycemia (1.2 ± 3.3 mg vs. 0.6 ± 1.7 mg in normoglycemia), and duration of GC treatment was longer (5.1 ± 2.1 days vs. 4.4 ± 2.2) in patients with GC-induced hyperglycemia (see also Table 1).

### 3.1. All Patients with GC-Induced Hyperglycemia vs. Normoglycemia

#### 3.1.1. Primary Endpoint

Seven hundred and fifty-seven patients (31%) experienced the primary endpoint, 292 (36%) with GC-induced hyperglycemia and 465 (29%) with normoglycemia (*p* < 0.001). In the unadjusted logistic regression model, odds ratio (OR) for the primary endpoint was 1.39 (95% CI 1.16–1.66) in the case of GC-induced hyperglycemia (see Table 3 for detailed results).

After adjusting for age, Charlson Comorbidity Index and GC dose, this effect increased to an OR of 1.68 (95% CI 1.25–2.26). Significant covariables were the Charlson Comorbidity Index (OR 1.09, 95% CI 1.04–1.14), percentage of glucose values in the range (OR 1.0, 95% CI 1.0–1.01), hypoglycemia (OR 1.95, 95% CI 1.2–3.17), and if indication for GCs were infectious or pneumological diseases, respectively (OR 1.39, 95% CI 1.04–1.87) or endocrine diseases (79% adrenal insufficiency or stress prophylaxis due to chronic GC use; OR 2.42, 95% CI 1.46–4.02; see also Table 4).

#### 3.1.2. Secondary Endpoints

Overall mortality was 7.4%. Sixty-six patients (8.1%) died in the GC-induced hyperglycemia group as compared to 113 (7.0%) in the normoglycemia group (*p* = 0.321).

In the unadjusted logistic regression model, mortality was not significantly different between groups (OR 1.17, 95% CI 0.86–1.61), and the results remained similar (OR 1.14, 95% 0.68–1.91) when adjusting for cofactors.

Significant covariables for mortality were age (OR 1.04, 95% CI 1.02–1.06, *p* < 0.0001), the Charlson Index (OR 1.26, 95% CI 1.17–1.35, *p* < 0.0001), and oncologic or endocrine indication for GCs (oncology: OR 2.46, 95% CI 1.25–4.85, *p* = 0.010; endocrinology: OR 2.72, 95% CI 1.09–6.78, *p* = 0.031).

Cardiovascular events occurred more often in patients with GC-induced hyperglycemia (*n* = 115, 14.2% vs. *n* = 162, 10.0% in normoglycemia, OR 1.48, 95% CI 1.14–1.91). In the adjusted model, this effect remained significant (OR 1.57, 95% CI 1.02–2.4).

Independent factors for cardiovascular events were age (OR 1.02, 95% CI 1.01–1.04, *p* = 0.003), the Charlson Index (OR 1.09, 95% CI 1.02–1.17, *p* = 0.015), and infectiology/pneumology indication for GCs (OR 1.55, 95% CI 1.04–2.32, *p* = 0.032).

The rate of infections was higher in patients with GC-induced hyperglycemia (*n* = 198, 24.3% vs. *n* = 301, 18.7% in normoglycemia, OR 1.40, 95% CI 1.15–1.72). This effect remained significant after adjusting for the above factors (OR 1.56, 95% CI 1.12–2.18). Independent factors for infections were the percentage of glucose readings in the range (OR 1.01, 95% CI 1.0–1.01, *p* = 0.045), hypoglycemia (OR 2.22, 95% CI 1.33–3.68, *p* = 0.002), endocrine indication for GCs (OR 1.82, 95% CI 1.04–3.18, *p* = 0.036), and mean glucose (OR 1.05, 95% CI 1.02–1.09, *p* = 0.006), For detailed results, see Table 3 and Table 4.

### 3.2. Patients with GC-Induced Diabetes vs. Normoglycemia

In the subgroup of patients with no pre-existing diabetes, the OR for reaching primary and secondary endpoints was similar to the overall cohort (see Table A3 for detailed results).

Significant covariables for reaching the primary endpoint were the Charlson Comorbidity Index (OR 1.12, 95% CI 1.05–1.19), hypoglycemia (OR 3.98, 95% CI 1.38–11.47), and endocrine indication for GCs (OR 2.8, 95% CI 1.47–5.37).

### 3.3. Patients with GC-Induced Diabetes vs. Pre-Existing Diabetes

There was no difference between the two groups for the primary combined endpoint (adjusted OR 0.98, 95% CI 0.68–1.40), for cardiovascular events (adjusted OR 0.98 (95% CI 0.6–1.6), and infections (adjusted OR 1.38, 95% CI 0.92–2.06). However, the risk of death was significantly lower in patients with pre-existing diabetes as compared to patients with GC-induced diabetes (OR 0.51, 95% CI 0.26–0.96, *p* = 0.043; see Table A4 for detailed results).

## 4. Discussion

In this single-center retrospective observational study, we found a rate of GC administration of 16% in in-hospital medical patients. In these patients, the incidence of GC-induced hyperglycemia added up to 33%, which is in accordance with the incidence found in the literature [3,4]. We observed an increased rate of mortality, cardiovascular events, and infections with an OR of 1.68 in GC-induced hyperglycemia compared to patients with normoglycemia, namely in patients without pre-existing diabetes. While especially cardiovascular events and infection rate were significantly increased, the non-significant difference in overall mortality might be due to a lack of power for this secondary endpoint or an absence of association.

The plausible mechanism by which short-term hyperglycemia might increase morbidity and mortality is by inducing endothelial dysfunction and oxidative stress. It has been shown that hyperglycemic excursions are associated with acute inflammation, endothelial dysfunction, and atherosclerotic plaque instability [23,29], even in the case of short-term hyperglycemia [30].

Notably, when comparing patients with new-onset GC-induced diabetes with hyperglycemic patients with pre-existing diabetes, the results were similar between the two groups. However, mortality was significantly lower in the diabetes group, even though the rate of hypoglycemia was higher. These results are supported by other studies, which showed less events in patients with diabetes than in patients without diabetes. Patients with pre-existing diabetes may benefit from higher glucose target ranges than those without diabetes [31,32].

Hypoglycemia is associated with overall mortality [33]. Hypoglycemia may trigger fatal cardiac arrhythmia, cerebral seizures, or stroke. In addition, increased adverse events in association with hypoglycemia may be a surrogate for more severe illness [14]. In our study, hypoglycemia was a strong and significant covariable for reaching the primary endpoint. For the secondary outcomes, including overall mortality, our study was most probably underpowered, albeit the results tended towards an elevated risk.

We detected an increased risk for adverse events for the vulnerable group of patients receiving GCs for stress prophylaxis, classified as endocrine diseases. This subgroup consisted of patients with primary, secondary, or tertiary adrenal insufficiency. Recently, it has been shown that the incidence and mortality of adrenal crisis has remained high and is potentially underestimated [34,35,36]. In this subgroup, GC administration, despite frequent side effects like hyperglycemia, is crucial for patient outcome.

There is a large potential impact of our results on clinical practice due to the number of affected patients and the extent of the effect in the hyperglycemia group [37]. The use of GCs has recently increased massively due to favorable data in patients with COVID-19 [2]. However, whether the outcome of GC-induced hyperglycemia is modifiable by the treatment of hyperglycemia is unknown. Treatment of GC-induced hyperglycemia should avoid both high glucose variability and hypoglycemia [12,31]. In patients with GC-induced hyperglycemia, there is an increased risk for high glucose variability. Keeping blood glucose within range is an extraordinary challenge, as GCs induce a pronounced postprandial hyperglycemia due to increased glucose intolerance and insulin resistance [38]. Furthermore, the risk for hypoglycemia, especially at nighttime, is elevated by the potential accumulation of daytime insulin, waning effect of glucocorticoids, and suppressed endogenous cortisol production [33,39]. As the vast majority of hypoglycemic events is induced by antidiabetic treatment, its occurrence should be avoided by all means, according to the “primum nil nocere” (first do no harm) principle [40].

So far, insulin treatment by an established hospital protocol with prandial and basal insulin, or premixed insulin is recommended [12]. Fast-acting insulins may increase glucose variability and bear a high risk of hypoglycemia. Antidiabetic agents like GLP-1 analogues or SGLT-2 inhibitors are promising alternatives to insulin as they have low risk for hypoglycemia, but the experience in GC-induced hyperglycemia is small, and again, outcome studies are lacking [41,42,43,44].

The main limitation of this study is its retrospective nature, which precludes causal reasoning. Furthermore, we cannot rule out having misclassified certain patients as not having pre-existing diabetes, even though diabetes was present but not yet diagnosed, which is partly attributable to not having HbA1c available on admission in all patients. Also, it was not possible to adjust for the severity of the illness leading to hospitalization due to the heterogeneity of the underlying diseases. The strongest factor of the combined primary endpoint was the high rate of infections. The observation time of 30 days might be considered too short, albeit there might be no visible effect beyond 30 days after short-term GC administration. The cohort was too small to be able to show a mortality benefit. Even though the diversity of underlying diseases might be considered a limitation, it has the advantage of being generalizable to a medical clinic population.

The strengths of this study are its large number of patients and the availability of 30-day interviews for outcome analysis. To the best of our knowledge, it is the first study with systematic reporting of outcome in GC-induced hyperglycemia.

## 5. Conclusions

The adverse events mortality, cardiovascular events, and infections were excessively higher in GC-induced hyperglycemia than in normoglycemia, and hypoglycemia was twofold higher in these patients. This effect was independent of pre-existing diabetes.

## Figures and Tables

**Figure 1 jcm-09-04079-f001:**
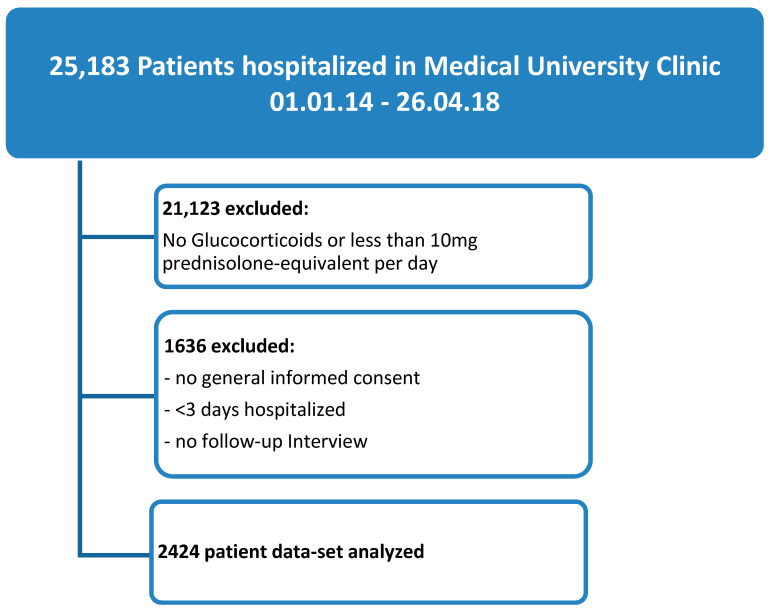
Study flow diagram.

**Table 1 jcm-09-04079-t001:** Baseline characteristics.

	All (*n* = 2424)	Patients with Hyperglycemia (*n* = 812)	Patients with Normoglycemia (*n* = 1612)	*p*-Value
Male sex, *n* (%)	1123 (46%)	356 (44%)	767 (47%)	0.084
Age, y	66.8 (±15.07)	68.5 (±13.4)	65.9 (±15.78)	<0.001
Body mass index, kg/m^2^	25.24 (22–30)	26.6 (23.4–31)	24.6 (21.5–28)	<0.001
Mean glucocorticoid (GC) dose ^1^	0.84 (±2.38)	1.22 (±3.3)	0.64 (±1.70)	<0.001
Duration of GC administration (in days)	4.64 (±2.19)	5.12 (±2.14)	4.4 (±2.17)	<0.001
Length of stay, days	8.0 (5.0–14)	9 (6–15)	8 (5–13)	<0.001
Indication for GC administration				<0.001
Autoimmune-inflammatory	816 (33.7%)	300 (36.9%)	516 (32%)	
Oncology	667 (27.5%)	175 (21.6%)	492 (30.5%)	
Pneumology/Infectiology	825 (34%)	294 (36.2%)	531 (32.9%)	
Endocrinology ^2^	116 (4.8%)	43 (5.3%)	73 (4.5%)	
Comorbidities (Charlson Index)	2.0 (1.0–4.0)	2.04 (1.0–3.0)	2.0 (1.0–4.0)	0.71
Diabetes-related parameters				
Glucose on admission	6.9 (5.8–8.6) mmol/L/124 (105–154) mg/dL	8.2 (6.5–10.6) mmol/L/148 (117–191) mg/dL	6.4 (5.6–7.5) mmol/L/115 (100–135) mg/dL	<0.001
Mean glucose differences from peaks to nadirs (MAGE)	3.75 (±2.14) mmol/L	4.89 (±2.18) mmol/L	2.40 (±1.01)	<0.001
Glucose (coefficient of variation) ^3^	0.25 (±0.10)	0.30 (±0.10)	0.20 (±0.07)	<0.001
Pre-existing diabetes	511 (21%)	413 (50%)	98 (6%)	<0.001
Insulin pre-treatment (including type 1 and type 3c/pancreatogenic)	185 (7.6%)	173 (21.3%)	12 (0.74%)	<0.001
Pre-treatment with glucose-lowering agents other than insulin	218 (8.9%)	178 (21.9%)	40 (2.48%)	<0.001
Diabetes treated with dietary measures	108 (4.5%)	62 (7.64%)	46 (2.85%)	<0.001

Data is shown as *n* (%) or mean/median (standard deviation or interquartile range) per group. ^1^ mean glucocorticoid GC dose is given as normalized GC dose in mg per prednisolone equivalent per kg per day, defined in Section 2.1. *Definitions and Categorizations*. ^2^ of which 92 (79%) had adrenal insufficiency or stress prophylaxis in chronic GC use, the remainder had hypercalcemia or amiodarone induced thyrotoxicosis, as defined in Section 2.1. *Definitions and Categorizations*. ^3^ coefficient of variation (CV) = SD/mean ∗ 100%. GC: glucocorticoid.

**Table 2 jcm-09-04079-t002:** Rate of hypoglycemia according to glycemia and diabetes status.

	Hypoglycemia, *n* (%)	No Hypoglycemia, *n* (%)	*p*-Value
All patients (*n* = 2424)	89 (3.7%)	2335 (96.3%)	
Patients with GC-induced hyperglycemia (*n* = 812)	78 (9.6%)	734 (90.4%)	
Patients with normoglycemia (*n* = 1612)	11 (0.7%)	1601 (99.3%)	<0.001
Patients without pre-existing diabetes (*n* = 1913)			
GC-induced diabetes (*n* = 399)	12 (3%)	387 (97%)	
Normoglycemia (*n* = 1514)	5 (0.3%)	1509 (99.7%)	<0.001
Patients with hyperglycemia (*n* = 812)			
pre-existing diabetes (*n* = 413)	66 (16%)	347 (84%)	
GC-induced diabetes (*n* = 399)	12 (3%)	387 (97%)	<0.001

Data is shown as *n* (%). GC: glucocorticoid. Patients with normoglycemia were mostly not treated with insulin, and their hypoglycemic glucose recordings were mostly fasting levels.

**Table 3 jcm-09-04079-t003:** Main results of primary and secondary endpoints in patients with and without hyperglycemia.

	All (*n* = 2424)	GC-Induced Hyperglycemia (*n* = 812)	Normoglycemia (*n* = 1612)	OR (95% CI)	*p*-Value	Adjusted OR ^1^ (95% CI)	*p*-Value
Primary Endpoint							
Combined endpoint: Death (30 days), cardiovascular events, and infections	757 (31%)	292 (36%)	465 (28.8%)	1.39 (1.16–1.66)	<0.001	1.68 (1.25–2.26)	0.001
Secondary Endpoints							
Death (30-days)	179 (7.4%)	66 (8.1%)	113 (7.0%)	1.17 (0.86–1.61)	0.321	1.13 (0.68–1.90)	0.63
Cardiovascular events	277 (11.4%)	115 (14.2%)	162 (10.0%)	1.48 (1.14–1.91)	0.003	1.56 (1.02–2.4)	0.040
Infections	498 (20.5%)	198 (24.3%)	301 (18.7%)	1.4 (1.15–1.72)	0.001	1.56 (1.12–2.18)	0.009

Data is shown as *n* (%) or mean/median (standard deviation or interquartile range) per group. ^1^ adjusted for age, pre-existing comorbidities by the Charlson Comorbidity Index [28], GC dose (mg prednisolone equivalent per kg body weight per GC day), indication for GCs, mean glucose, CV of glucose, percent of glucose values in the range, and hypoglycemia < 4.0 mmol/L (<72 mg/dL). GC: glucocorticoid.

**Table 4 jcm-09-04079-t004:** Detailed results of the multivariable logistic regression model for the primary and secondary endpoints.

Factor	OR	95% CI	*p*-Value
Hyperglycemia	1.68	1.25–2.26	0.001
Age	1.01	0.99–1.02	0.069
Charlson Comorbidity Index	1.09	1.04–1.14	<0.001
Coefficient of glucose variation	0.46	0.11–2.0	0.3
Percent of glucose in range	1.01	1.0–1.01	0.02
Hypoglycemia	1.95	1.20–3.17	0.007
Indication for GC administration			
Oncology	1.35	0.92–1.97	0.122
Pneumology/Infectiology	1.39	1.04–1.87	0.027
Endocrinology ^1^	2.42	1.46–4.02	0.001
Mean glucose	1.03	1.0–1.07	0.050
GC dose ^2^	0.98	0.93–1.05	0.59

For the primary endpoint in patients with hyperglycemia, detailed results for the cofactors are shown. GC: Glucocorticoids. ^1^ of which 92 (79%) had adrenal insufficiency or stress prophylaxis in chronic GC use. ^2^ mean GC dose is given as normalized GC dose in mg per prednisolone equivalent per kg per day.

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
