# Peer review of "Outcomes of Hospitalized Patients with Glucocorticoid-Induced Hyperglycemia—A Retrospective Analysis"

_jcm, 2020, doi:10.3390/jcm9124079_

Round 1

Reviewer 1 Report

The present paper copes with a really interesting topic, that is usually poorly investigated or at least investigated on small numbers. Authors did a great job by reviewing a high number of medical records.

Some concerns are raised by this reviewer regarding the methodology and the general clarity of the paper. Authors are recommended to consider the comments listed below.

  • What guidelines are you referring as to in the Introduction? Please include a reference or just report findings by van der Berghe alone.
  • Tight glucose control was observed to be harmful in the ACCORD trial and it is even more harmful in those with a concurrent criticall illness. The impact on the CV system is deleterious and the risk of mortality is increased, as outlined in these reviews ( Lancet Diabetes Endocrinol. 2019 May;7(5):385-396).
  • How were the criteria to define steroid-induced hyperglycemia fixed? A couple of reviews on this topic might be of help, since the diagnosis of this entity is still a matter of debate (World J Diabetes. Jul 25, 2015; 6(8): 1073-1081; Diabetes Res Clin Pract. 2018 May;139:203-220).
  • I would like authors to calculate also MAGE as an index for glycemic variability in order to catch Major glucose fluctuations, as suggested by Ceriello et al., Lancet Diabetes Endocrinol2019; 7: 221–30. This then needs to be detailed in the Methods section.
  • Is this a composite endpoint "30-day mortality, cardiovascular events, and infections" or are these entities considered as single items? I assume yes, please detail/clarify this aspect. Similarly, were secondary outcomes considered separately or all together?
  • Paragraph 2.3 appears to be of poor help. You may eventually want to provide a sample size in order to evaluate whether your findings might be considered reliable or not, although the observational design has known limitation.
  • Did you test continous variables for normality? If not, please do it and add this in the manuscript.
  • What does this mean "We performed complete case analyses only"?
  • How were variables included in the adjusted models chosen? Please detail this aspect, which is really challenging and influences all of your analyses.
  • Data included at the bottom of table 1 must be included within the manuscript.
  • Table 1: the division between "with and without hyperglycemia" refers as to hyperglycemia due to causes other than steroid-induced hyperglycemia? If yes, this needs to be well detailed. In addition, if this is confirmed, a doubt on the presence of an unknown diabetes arises. Please revise this table and the data accordingly.
  • Was hypoglycemia spontaneous or iatrogenic, as I suspect? Please address.
  • As you say that "Mean GC dose (prednisolone equivalents/kg/day) was higher in patients with GC-induced hyperglycemia (1.2 ± 3.3 mg vs. 0.6 ± 1.7 mg in normoglycemia), whereas duration of GC treatment was shorter (4.4 ± 2.2 days vs. 5.1 ± 2.1) in patients with GC-induced hyperglycemia (see also Table 1)", did you find any correlation between the dose of glucocorticoids and the extent of hyperglycemia? Any inverse correlation between glucocorticoid treatment duration and the extent of hyperglycemia?
  • I would suggest to put together tables 3 and 4, this will help readers better understand the results now included in table 3.
  • Not clear to me the outcome CV events in table 3, lower events in the GC-induced hyperglycemia but higher risk? Is this correct? Numbers in the table and in the text are different, as you mentioned what I suspected in the text, while in the table is different. Please address this point.
  • Not clear what "(new) GC-induced diabetes vs. preexisting diabetes" refers to. Did you include patients with prior GC-induced diabetes in this cohort? Although the message provided in paragraph 3.3 is important, I feel it is not easy to follow.
  • Discussion should be revised. Are you really sure that short-term hyperglycemia may immediately lead to endothelial dysfunction and oxidative stress? Is this something happening so fast? Or is this a process needing some time to take place. It is important also to consider the duration of GC therapy when you declare this.

MINOR POINTS

  • Please provide glycemic values also in mg/dL for increased clarity.
  • Probably steroid-induced hyperglycemia is more advisable compared with GC-induced hyperglycemia.
  • "congestive heart failure" --> for this do you mean acute decompensated heart failure needing hospital admission? Please detail.
  • Paragraph 2.2 might be simplified to Endpoints only.
  • Charlson Index not described in the Methods. Please address this point and detail whether you use age-adjusted CCI or not.
  • "pancreatogenic" diabetes is no longer used. What are you meaning with this?
  • Please avoid the adjective "diabetic" and change it with "with diabetes".
  • This reference may be worth being cited: Ann Nutr Metab 2014;65:324–33. I suggested some other ones, more updated in the topic, in other comments.

Author Response

Dear Ms. Ren,

Dear editors,

Dear reviewers,

We thank you for the opportunity to resubmit our manuscript “Outcomes of hospitalized patients with glucocorticoid-induced hyperglycemia – a retrospective analysis”.

We have endeavoured to respond as constructively and as thoroughly as possible to all your reviewers’ comments and hope that our revised manuscript will prove acceptable.

Please find our detailed point-to-point answers to the issues raised below. The changes made in the revised manuscript itself are highlighted using the “track-changes” function, or if not displayed well this way, highlighted in yellow. The line numbering of our point-to-point-answers refer to the manuscript with track changes.

As there were some changes in the references, we have used the version with the endnote references in order to be able to provide correct reference output.

Sincerely yours,

Claudine Blum

Reviewer 1

The present paper copes with a really interesting topic, that is usually poorly investigated or at least investigated on small numbers. Authors did a great job by reviewing a high number of medical records.

We thank the reviewer for this positive comment.

Some concerns are raised by this reviewer regarding the methodology and the general clarity of the paper. Authors are recommended to consider the comments listed below.

  • What guidelines are you referring as to in the Introduction? Please include a reference or just report findings by van der Berghe alone.

We have inserted the reference of Umpierrez et al. (line 45).[1]

  • Tight glucose control was observed to be harmful in the ACCORD trial and it is even more harmful in those with a concurrent criticall illness. The impact on the CV system is deleterious and the risk of mortality is increased, as outlined in these reviews ( Lancet Diabetes Endocrinol. 2019 May;7(5):385-396).

We thank the reviewer for this reference suggestion and added it to our introduction (line 51).

  • How were the criteria to define steroid-induced hyperglycemia fixed? A couple of reviews on this topic might be of help, since the diagnosis of this entity is still a matter of debate (World J Diabetes. Jul 25, 2015; 6(8): 1073-1081; Diabetes Res Clin Pract. 2018 May;139:203-220).

We took the definition of hyperglycemia from the guideline paper of Umpierrez et al[1], where the case of GC-induced hyperglycemia is described as well. This definition is supported by the American Diabetes Association (ADA) and widely used.[2,3]

For easier comprehension, we have modified the definition slightly as follows: “GC – induced hyperglycemia was defined as either morning fasting blood glucose of > 7.0 mmol/l (126 mg/dL) or a random glucose measurement of >11.0 mmol/l (198 mg/dL) after start of GCs, regardless of pre-existing diabetes status. Normoglycemia was defined as fasting blood glucose of 7.0 mmol/l (126 mg/dL) or lower and random glucose of 11.0 mmol/l (198 mg/dL) or lower.[1,3,4] GC-induced diabetes was defined if the above criteria for hyperglycemia were met with no pre-existing diabetes.” (line 78ff.)

  • I would like authors to calculate also MAGE as an index for glycemic variability in order to catch Major glucose fluctuations, as suggested by Ceriello et al., Lancet Diabetes Endocrinol2019; 7: 221–30. This then needs to be detailed in the Methods section.

We have now calculated mean differences from peaks to nadirs (MAGE); see Baseline Table 1. We mentioned this in the methods section as well (line 90).

  • Is this a composite endpoint "30-day mortality, cardiovascular events, and infections" or are these entities considered as single items? I assume yes, please detail/clarify this aspect. Similarly, were secondary outcomes considered separately or all together?

The composite primary endpoint consisted of 30-day mortality, cardiovascular events, and infections. Each of the single items of the combined endpoint were also analyzed separately as a secondary endpoint. We have clarified this point further as follows: “The objective of this study was to evaluate the outcome in hospitalized patients with GC- induced hyperglycemia. The primary outcome was the combined endpoint of 30-day mortality, cardiovascular events, and infections. Secondary endpoints were each of the single items of the composite endpoint: 30-day mortality, 30-day cardiovascular events, 30-day infection, and in addition in-hospital hypoglycemia.” (lines 105ff.)

  • Paragraph 2.3 appears to be of poor help. You may eventually want to provide a sample size in order to evaluate whether your findings might be considered reliable or not, although the observational design has known limitation.

Thank you for your valuable input. Sample size/power calculations for retrospective studies are a highly debated topic, but we can understand the wish to understand the strength of the current results to previous findings. As such, we have performed a power calculation based on the findings of a Systematic Review and Meta-Analysis on glycemic control in non-critically ill patients by Murad et al. [5]. Pooled relative risk ratios for death, infection, myocardial infarction, and stroke were 0.85 (95% CI 0.58-1.26), 0.41 (95% CI 0.21-0.77), 0.69 (95% CI 0.37-1.28), and 0.63 (95% CI 0.29-1.38), respectively. Based on these findings, we decided to pick one of the more conservative ratios and assumed the overall effect size similar to the one for myocardial infarction. Assuming an alpha level of 5%, an intervention group size of 812 with a rate ratio of 0.02 and a control group size of 1,612 with a rate ratio of 0.039 (numbers based on figure 2 from Murad et al.), a Chi-squared test on two proportions gives a power of 0.73. Hence, we have added the following lines to paragraph 2.3 (lines 114 ff.):

“A Systematic Review and Meta-Analysis on glycemic control in non-critically ill patients by Murad et al. [5] found pooled relative risk ratios for death, infection, myocardial infarction, and stroke of 0.85 (95% CI 0.58-1.26), 0.41 (95% CI 0.21-0.77), 0.69 (95% CI 0.37-1.28), and 0.63 (95% CI 0.29-1.38), respectively. Based on their findings, we performed a Chi-squared two proportions power calculation assuming an alpha level of 5%, an intervention group size of 812 with a rate ratio of 0.02 and a control group size of 1,612 with a rate ratio of 0.039 (numbers based on figure 2 from Murad et al.) giving a power of 0.73.”

  • Did you test continuous variables for normality? If not, please do it and add this in the manuscript.

Thank you for your remark. Test for normality such as Shapiro-Wilks test have their pitfalls. Whereas they lack power in case of a small sample, they are overly powerful in large samples as it is in ours. Thus, they tend to give false-positive results. Therefore, we abstained using them. Instead, we have checked distribution graphically by histograms and quantile-quantile plots. Furthermore, in large samples adding independent variables to a statistical model their normalized sum tends towards a normal distribution (“central limit theorem”). Albeit the actual size needed is a matter of debate, mostly a sample size larger than 50 to 100 per group is regarded as sufficient. As such, we regarded our sample as large enough to invoke the central limit theorem. To clarify this, we have added the following sentence to our methods section: ”We did not use test for normality such as the Shapiro-Wilks test, due to their tendency to give false-positive results in large samples. Instead, we checked graphically by plotting histograms and quantile-quantile plots. Discrete variables are expressed as counts (percentages) and continuous variables as either means (SD) if normally distributed or medians (IQR) if non-normally distributed.” (see lines 123ff.)

  • What does this mean "We performed complete case analyses only"?

We have rephrased the sentence to: “We have only performed analysis with complete cases. Thus, we did not perform any imputation method for missing covariates and cases with missing covariates were excluded from the multivariable analysis.” (lines 129 ff.)

  • How were variables included in the adjusted models chosen? Please detail this aspect, which is really challenging and influences all of your analyses.

We agree that there is always a debate how statistical models should be adjusted to potential confounders. Principally, there are two options, a data-driven approach based on effect size/p-value or an approached based on expert opinion. We had the impression that a data driven approach would not fit our data and would introduce bias. As such, we have decided to adjust our model for age, Charlson Comorbidity Index, GC dose, indication for GCs, mean glucose, CV of glucose, percent of glucose values in range, and hypoglycemia <4.0 mmol/l. This decision was based on expert opinion by the study authors as endocrinologists/diabetologists. Also, we have reviewed the literature and found the above variables as potential confounders. To reflect this process, we have added the following sentence to the methods section: “Choice of adjusted variables was based on expert opinion and review of the literature. The variables chosen were identified by previous research as important confounders.[6-9]” (see also lines 146 ff.)

  • Data included at the bottom of table 1 must be included within the manuscript.

Although all data at the bottom of table 1 is included within the manuscript, we agree that this can be misinterpreted. Hence, we rephrased this passage accordingly, pointing to the corresponding sections in the manuscript:” 1 mean GC dose is given as normalized GC dose in mg per prednisolone equivalent per kg per day, defined in section 2.1. Definitions and Categorizations 2of which 92 (79 %) had adrenal insufficiency or stress prophylaxis in chronic GC use, the remainder had hypercalcemia or amiodarone induced thyrotoxicosis, , defined in section 2.1. Definitions and Categorizations”.

For quick reference, the GC dose is defined in lines 93ff.

The endocrinology group is defined in line 102ff.

  • Table 1: the division between "with and without hyperglycemia" refers as to hyperglycemia due to causes other than steroid-induced hyperglycemia? If yes, this needs to be well detailed. In addition, if this is confirmed, a doubt on the presence of an unknown diabetes arises. Please revise this table and the data accordingly.

In table 1, we compared baseline characteristics of patients receiving glucocorticoids who had subsequent hyperglycemia to patients receiving glucocorticoids who did not develop hyperglycemia at all.

We agree that the column headers can be misinterpreted. Thus, we have rephrased the headers to “Patients with hyperglycemia” and “Patients with normoglycemia” to remove any ambiguity.

The percentages of the diabetes status in the different groups are presented in Table 1.

  • Was hypoglycemia spontaneous or iatrogenic, as I suspect? Please address.

As all patients with hyperglycemia received insulin treatment, hypoglycemic events in this group are per definition iatrogenic. The very few events (see table 2) in the group with normoglycemia and no diabetes were most probably due to fasting and thus spontaneous, although we did not review the charts of these cases separately.

We have commented on this point in the discussion on lines 325ff., as follows: “As the vast majority of hypoglycemic events is induced by antidiabetic treatment, its occurrence should be avoided by all means, according to the “primum nil nocere – first do no harm” principle.[10]

  • As you say that "Mean GC dose (prednisolone equivalents/kg/day) was higher in patients with GC-induced hyperglycemia (1.2 ± 3.3 mg vs. 0.6 ± 1.7 mg in normoglycemia), whereas duration of GC treatment was shorter (4.4 ± 2.2 days vs. 5.1 ± 2.1) in patients with GC-induced hyperglycemia (see also Table 1)", did you find any correlation between the dose of glucocorticoids and the extent of hyperglycemia? Any inverse correlation between glucocorticoid treatment duration and the extent of hyperglycemia?

We apologize and thank the reviewer for noticing this mistake. The columns have been switched erroneously. We have corrected the according numbers in Table 1. The duration of GC- treatment was indeed longer in the group with GC- induced hyperglycemia.

In regression analysis, we found no association between mean GC dose and our primary outcome (table 3b, OR 0.98, 95% CI 0.93-1.05, p=0.59). Further, percentage of glucose readings in range was a significant covariate indicating that hyperglycemia might be relevant for the outcome. Primarily, we were interested in clinical outcome and thus did not investigate the extent of hyperglycemia.

  • I would suggest to put together tables 3 and 4, this will help readers better understand the results now included in table 3.

We have put the former tables 3 and 4 together as table 3a and table 3b for better understanding.We have also added the note that the values in table 3b are referring for the model of our primary endpoint.

  • Not clear to me the outcome CV events in table 3, lower events in the GC-induced hyperglycemia but higher risk? Is this correct? Numbers in the table and in the text are different, as you mentioned what I suspected in the text, while in the table is different. Please address this point.

We apologize for this error and are obliged to the reviewer for noticing. The numbers in the text were correct, but not in the table. We have corrected the according numbers in table 3a.

  • Not clear what "(new) GC-induced diabetes vs. preexisting diabetes" refers to. Did you include patients with prior GC-induced diabetes in this cohort? Although the message provided in paragraph 3.3 is important, I feel it is not easy to follow.

As we have specified in the methods section under 2.1. Definitions and Categorizations, GC-induced diabetes was defined if the criteria for hyperglycemia were met and the patient had no pre-existing diabetes (lines 78ff.). We reassure that these patients did not have a previous diabetes diagnosis.

To make this point clearer in the results section, we have delete the term (new) in the title: 3.3. Patients with (new) GC-induced diabetes vs. preexisting diabetes (line 257). Furthermore, we have replaced the term “new-onset GC-induced diabetes” with “GC-induced diabetes” throughout the manuscript.

  • Discussion should be revised. Are you really sure that short-term hyperglycemia may immediately lead to endothelial dysfunction and oxidative stress? Is this something happening so fast? Or is this a process needing some time to take place. It is important also to consider the duration of GC therapy when you declare this.

Thank you for this important question. There is indeed evidence that even short-term hyper- and hypoglycemia may induce endothelial dysfunction. In glucose clamp studies in healthy individuals, this effect has been shown.[11] We have added this fact to the discussion (line 295).

MINOR POINTS

  • Please provide glycemic values also in mg/dL for increased clarity.

We now give all glycemic values in both mmol/l and mg/dL.

  • Probably steroid-induced hyperglycemia is more advisable compared with GC-induced hyperglycemia.

Thank you for this input. We believe that the term glucocorticoid should be used, in order to be as precise as possible – steroid is much more a general term for a larger group of substances. Furthermore, the term glucocorticoid explicitly underlines the effect which is central to this manuscript. We would like to leave it to the editors to decide whether steroid or glucocorticoid or corticosteroid should be used, as usually there is an overall journal policy of the use of specific terms.

  • "congestive heart failure" --> for this do you mean acute decompensated heart failure needing hospital admission? Please detail.

Yes, we mean acute decompensated heart failure needing hospital admission. We have changed this term accordingly (line 85).

  • Paragraph 2.2 might be simplified to Endpoints only.

We have now simplified the title and changed the paragraph as follows (lines 104ff.):

“2.2. Endpoints

The objective of this study was to evaluate the outcome in hospitalized patients with GC- induced hyperglycemia. The primary endpoint was the combined endpoint of 30-day mortality, cardiovascular events, and infections. Secondary endpoints were each of the single items of the composite endpoint: 30-day mortality, 30-day cardiovascular events, 30-day infections,; and in addition, in-hospital hypoglycemia.”

  • Charlson Index not described in the Methods. Please address this point and detail whether you use age-adjusted CCI or not.

We used the age-adjusted CCI in order to adjust for age and CCI. The Charlson Comorbidity Index is now introduced more in detail in the methods section in lines 144ff. as follows: “ Then, these three models were adjusted for age and preexisting comorbidities by age-adjusted Charlson Comorbidity Index,[12] by GC dose (mg prednisolone equivalent per kg body weight per GC day), reasons for GCs, mean glucose, CV of glucose, percent of glucose values in range, and hypoglycemia <4.0 mmol/l (< 72 mg/dL)». See also factors of the multivariable logistic regression model in table 3b.

  • "pancreatogenic" diabetes is no longer used. What are you meaning with this?

By pancreatogenic diabetes, we mean diabetes type 3c, i.e. diabetes secondary to removal of the pancreas or secondary to a pancreatic disease, such as chronic pancreatitis or pancreatic cancer. We have added the term type 3c as well.(table 1).

  • Please avoid the adjective "diabetic" and change it with "with diabetes".

We apologize for this lapse. We have removed the term “diabetic“ accordingly.

  • This reference may be worth being cited: Ann Nutr Metab 2014;65:324–33. I suggested some other ones, more updated in the topic, in other comments.

We thank for this suggestion and have now added it in the introduction (line 43).

References

  1. Umpierrez, G.E.; Hellman, R.; Korytkowski, M.T.; Kosiborod, M.; Maynard, G.A.; Montori, V.M.; Seley, J.J.; Van den Berghe, G.; Endocrine, S. Management of hyperglycemia in hospitalized patients in non-critical care setting: an endocrine society clinical practice guideline. J Clin Endocrinol Metab 2012, 97, 16-38, doi:10.1210/jc.2011-2098.
  2. The International Classification of Headache Disorders: 2nd edition. Cephalalgia 2004, 24 Suppl 1, 9-160.
  3. Suh, S.; Park, M.K. Glucocorticoid-Induced Diabetes Mellitus: An Important but Overlooked Problem. Endocrinol Metab (Seoul) 2017, 32, 180-189, doi:10.3803/EnM.2017.32.2.180.
  4. American Diabetes, A. 15. Diabetes Care in the Hospital: Standards of Medical Care in Diabetes-2019. Diabetes Care 2019, 42, S173-S181, doi:10.2337/dc19-S015.
  5. Murad, M.H.; Coburn, J.A.; Coto-Yglesias, F.; Dzyubak, S.; Hazem, A.; Lane, M.A.; Prokop, L.J.; Montori, V.M. Glycemic control in non-critically ill hospitalized patients: a systematic review and meta-analysis. J Clin Endocrinol Metab 2012, 97, 49-58, doi:10.1210/jc.2011-2100.
  6. Blackburn, D.; Hux, J.; Mamdani, M. Quantification of the Risk of Corticosteroid-induced Diabetes Mellitus Among the Elderly. J Gen Intern Med 2002, 17, 717-720.
  7. Gulliford, M.C.; Charlton, J.; Latinovic, R. Risk of diabetes associated with prescribed glucocorticoids in a large population. Diabetes Care 2006, 29, 2728-2729, doi:10.2337/dc06-1499.
  8. Hwang, J.L.; Weiss, R.E. Steroid-induced diabetes: a clinical and molecular approach to understanding and treatment. Diabetes Metab Res Rev 2014, 30, 96-102, doi:10.1002/dmrr.2486.
  9. Liu, X.X.; Zhu, X.M.; Miao, Q.; Ye, H.Y.; Zhang, Z.Y.; Li, Y.M. Hyperglycemia induced by glucocorticoids in nondiabetic patients: a meta-analysis. Annals of nutrition & metabolism 2014, 65, 324-332, doi:10.1159/000365892.
  10. Mealey, B.L. Commentary: managing patients with diabetes: first, do no harm. J Periodontol 2007, 78, 2072-2076, doi:10.1902/jop.2007.070362.
  11. Joy, N.G.; Perkins, J.M.; Mikeladze, M.; Younk, L.; Tate, D.B.; Davis, S.N. Comparative effects of acute hypoglycemia and hyperglycemia on pro-atherothrombotic biomarkers and endothelial function in non-diabetic humans. Journal of diabetes and its complications 2016, 30, 1275-1281, doi:10.1016/j.jdiacomp.2016.06.030.
  12. Stolz, D.; Christ-Crain, M.; Morgenthaler, N.G.; Miedinger, D.; Leuppi, J.; Muller, C.; Bingisser, R.; Struck, J.; Muller, B.; Tamm, M. Plasma pro-adrenomedullin but not plasma pro-endothelin predicts survival in exacerbations of COPD. Chest 2008, 134, 263-272.
  13. Wallace, M.D.; Metzger, N.L. Optimizing the Treatment of Steroid-Induced Hyperglycemia. Ann Pharmacother 2018, 52, 86-90, doi:10.1177/1060028017728297.
  14. Radhakutty, A.; Burt, M.G. MANAGEMENT OF ENDOCRINE DISEASE: Critical review of the evidence underlying management of glucocorticoid-induced hyperglycaemia. Eur J Endocrinol 2018, 10.1530/EJE-18-0315, doi:10.1530/EJE-18-0315.
  15. Schuetz, P.; Hausfater, P.; Amin, D.; Haubitz, S.; Fassler, L.; Grolimund, E.; Kutz, A.; Schild, U.; Caldara, Z.; Regez, K., et al. Optimizing triage and hospitalization in adult general medical emergency patients: the triage project. BMC emergency medicine 2013, 13, 12, doi:10.1186/1471-227X-13-12.
  16. Ceriello, A.; Esposito, K.; Piconi, L.; Ihnat, M.A.; Thorpe, J.E.; Testa, R.; Boemi, M.; Giugliano, D. Oscillating glucose is more deleterious to endothelial function and oxidative stress than mean glucose in normal and type 2 diabetic patients. Diabetes 2008, 57, 1349-1354, doi:10.2337/db08-0063.
  17. Ceriello, A.; Monnier, L.; Owens, D. Glycaemic variability in diabetes: clinical and therapeutic implications. The lancet. Diabetes & endocrinology 2019, 7, 221-230, doi:10.1016/S2213-8587(18)30136-0.

Reviewer 2 Report

Introduction

1) Line 52: there is mention of few studies of studies on treatment for GC-induced hyperglycemia, but the referenced studies - 9 and 14 - do not include these studies. 

Examples include:

Myers AK, Khan M, Choi S, Garnica P, Stoffels G, Lin A. Implementation of a Weight-Based Protocol for the Management of Steroid-Induced Hyperglycemia. American Journal of Therapeutics. 2020 Jul 1;27(4):e392-9.

Suh S, Park MK. Glucocorticoid-induced diabetes mellitus: an important but overlooked problem. Endocrinol Metab. 2017;32:180–189.   Kwon S, Hermayer KL, Hermayer K. Glucocorticoid-induced hyperglycemia. Am J Med Sci. 2013;345:274–277.   Wallace MD, Metzger NL. Optimizing the treatment of steroid-induced hyperglycemia. Ann Pharmacother. 2018;52:86–90.   Dhital SM, Shenker Y, Meredith M, et al. A retrospective study comparing neutral protamine Hagedorn insulin with glargine as basal therapy in prednisone-associated diabetes mellitus in hospitalized patients. Endocr Pract. 2012;18:712–719.   Methods 1) Why did you only examine 30 days? Any consideration for 90 days as well. 2) The composite primary outcome is very confusing. Why not simply focus on mortality? Infection is confounding as it is not clear if the infection occurred before or after GC use.  3) The persons used was a very heterogenous mix of people. I think that it would have been better to remove the people who had chronic AI, as they require stress-dosing which is different than what is required for persons with a COPD exacerbation or those getting cancer treatment. 4) Why was prednisone 10mg used as a cutoff. 5mg is physiological dosing so for patients on 7.5mg, why not include them. 5) It would have been helpful to stratify patients by their prednisone dosing as I would expect far more hyperglycemia with prednisone dosed at 1mg/kg vs 10mg.   Results 1) When discussing the overall mortality, it would have been helpful to know the cause of death in order to understand if the GC-induced hyperglycemia may have played a role.  2) Again, unclear if infections started before or after the initiation of steroids.  3) Sentence 236-238 in section 3.3.2 is unclear, can you clarify this as its an important point.    Discussion 1) You mention oral agents including SGLT2i and GLP1RA, but you may want to had oral or sq.  2) You mention that the diversity of of disease may have impacted your results but you did not account for this in the data analysis.    Overall 1) It is not good to label people by their disease so terming persons as non-diabetic or diabetic is not appropriate. This was done in multiple parts of the paper.  2) Other papers have shown that hyperglycemia predicts mortality in those with and without diabetes, but you did not comment on why your data did not show this.

Author Response

Revision – jcm-996130                                              Aarau, November 30, 2020

Dear editors,

Dear reviewers,

We thank you for the opportunity to resubmit our manuscript “Outcomes of hospitalized patients with glucocorticoid-induced hyperglycemia – a retrospective analysis”.

We have endeavoured to respond as constructively and as thoroughly as possible to all your reviewers’ comments and hope that our revised manuscript will prove acceptable.

Please find our detailed point-to-point answers to the issues raised below. The changes made in the revised manuscript itself are highlighted using the “track-changes” function, or if not displayed well this way, highlighted in yellow. The line numbering of our point-to-point-answers refer to the manuscript with track changes.

As there were some changes in the references, we have used the version with the endnote references in order to be able to provide correct reference output.

Sincerely yours,

Claudine Blum

Reviewer 2

Introduction

1) Line 52: there is mention of few studies of studies on treatment for GC-induced hyperglycemia, but the referenced studies - 9 and 14 - do not include these studies. 

Examples include:

Myers AK, Khan M, Choi S, Garnica P, Stoffels G, Lin A. Implementation of a Weight-Based Protocol for the Management of Steroid-Induced Hyperglycemia. American Journal of Therapeutics. 2020 Jul 1;27(4):e392-9.

Suh S, Park MK. Glucocorticoid-induced diabetes mellitus: an important but overlooked problem. Endocrinol Metab. 2017;32:180–189.    

Kwon S, Hermayer KL, Hermayer K. Glucocorticoid-induced hyperglycemia. Am J Med Sci. 2013;345:274–277.  

Wallace MD, Metzger NL. Optimizing the treatment of steroid-induced hyperglycemia. Ann Pharmacother. 2018;52:86–90.  

Dhital SM, Shenker Y, Meredith M, et al. A retrospective study comparing neutral protamine Hagedorn insulin with glargine as basal therapy in prednisone-associated diabetes mellitus in hospitalized patients. Endocr Pract. 2012;18:712–719.  

We thank the reviewer for this input. We have cited the two systematic reviews[13,14] and the latest expert guidelines,[4] which list all the studies investigating treatment of GC-induced hyperglycemia, but refrained on purpose from listing each study separately, as well as narrative reviews, expert opinions, or articles on the incidence of GC-induced hyperglycemia without connection to its treatment. We have of course cited supporting studies where appropriate. As this is not a literature review, we had to make such limitiations, as the number of allowed references is limited.

Methods 1) Why did you only examine 30 days? Any consideration for 90 days as well.

We understand the interest in 90-day outcome. For this retrospective analysis, only 30-day follow-up interviews were available. This was the only time point when follow-up was performed.

2) The composite primary outcome is very confusing. Why not simply focus on mortality? Infection is confounding as it is not clear if the infection occurred before or after GC use. 

We have looked at mortality as a secondary endpoint, but the number of events was too low for mortality only. We have assured that all recorded infections occurred after GC use by reviewing each case and its consecutive adverse events meticulously through chart review by two independent persons.

3) The persons used was a very heterogenous mix of people. I think that it would have been better to remove the people who had chronic AI, as they require stress-dosing which is different than what is required for persons with a COPD exacerbation or those getting cancer treatment.

In this cohort, the number of patients would have been too small for any analysis, had we formed smaller groups. In order to limit confounding, we adjusted for indication of GCs such as stress-dosing versus COPD exacerbation versus cancer treatment and for GC dose. Thus we are confident that our results are robust, as we believe we have considered all relevant confounders.

4) Why was prednisone 10mg used as a cutoff. 5mg is physiological dosing so for patients on 7.5mg, why not include them.

Our goal was to analyze whether the in-hospital treatment with GC for acute conditions had any adverse effects. Therefore, we excluded patients on a minimal chronic GC dose not receiving any additional GCs at all, as this usually reflects a chronic condition. We wanted to eliminate patients with “physiological” dosing of GCs, but include patients who would have a certain risk of developing hyperglycemia during the short time of hospital admission.

5) It would have been helpful to stratify patients by their prednisone dosing as I would expect far more hyperglycemia with prednisone dosed at 1mg/kg vs 10mg.  

Thank you for this valuable input. We agree that there is abundant evidence that higher dosing of glucocorticoids leads to more hyperglycemia.[8,9] Although stratification or factorization of variables might increase interpretability, in our case it made the statistical models unstable. Also, we have decided to increase the power of our analysis by keeping as many variables as possible continuous. Our decision is supported by the fact that the cumulative GC dose was not a significant factor in the multivariate analysis (OR 0.98, 95%CI 0.93-1.05, p=0.59; see also Table 3b).

Results 1) When discussing the overall mortality, it would have been helpful to know the cause of death in order to understand if the GC-induced hyperglycemia may have played a role. 

We agree with you that this piece of information might be very insightful. Unfortunately, telephone follow-up was conducted as part of another study at our institution. From our experience, the information on cause of death was often not helpful, as the indication given by the contacted next of kin were not reliable, and a definitive statement from a physician was only available if patients died in-hospital.

2) Again, unclear if infections started before or after the initiation of steroids. 

Again, we assure the reviewer that endpoints such as infections had to be met after the initiation of glucocorticoids. Incidence of outcomes was done individually by two independent persons doing chart review in all patients.

3) Sentence 236-238 in section 3.3.2 is unclear, can you clarify this as its an important point.   

We have now rewritten and shortened the paragraph 3.3. as follows (lines 257 ff.):

“3.3. Patients with GC-induced diabetes vs. pre-existing diabetes

There was no difference between the two groups for the primary combined endpoint (adjusted OR 0.98, 95% CI 0.68-1.40), for cardiovascular events (adjusted OR 0.98 (95% CI 0.6-1.6), and infections (adjusted OR 1.38, 95% CI 0.92-2.06). However, the risk of death was significantly lower in patients with pre-existing diabetes as compared to patients with GC-induced-diabetes (OR 0.51, 95% CI 0.26-0.96, p=0.043; see Table 4A for detailed results).

Discussion 1) You mention oral agents including SGLT2i and GLP1RA, but you may want to had oral or sq. 

Thank you for being precise on this point. We have deleted the term oral antidiabetic agents and now only state antidiabetic agents, as there are now s.c. and oral GLP-1 analogues available ( line 330).

2) You mention that the diversity of disease may have impacted your results but you did not account for this in the data analysis.   

We decided to account for the diversity of disease by adjusting for the indication of GCs and by adjusting for comorbidity burden according to Charlson index. However, we admit that in doing so, we might not account for other factors which may be inherent to the underlying diseases themselves.

 Overall 1) It is not good to label people by their disease so terming persons as non-diabetic or diabetic is not appropriate. This was done in multiple parts of the paper. 

We apologize for this lapse and have now removed the term “diabetic” throughout the manuscript.

2) Other papers have shown that hyperglycemia predicts mortality in those with and without diabetes, but you did not comment on why your data did not show this.

Our sample size is not powered to show a mortality difference. The sample size would have to be at least 4000 patients to show such a difference. We now mention this in the limitations section (line 341). See also the paragraph on sample size/power calculation added to 2.3. (lines 114ff):

“A Systematic Review and Meta-Analysis on glycemic control in non-critically ill patients by Murad et al. [5] found pooled relative risk ratios for death, infection, myocardial infarction, and stroke of 0.85 (95% CI 0.58-1.26), 0.41 (95% CI 0.21-0.77), 0.69 (95% CI 0.37-1.28), and 0.63 (95% CI 0.29-1.38), respectively. Based on their findings, we performed a Chi-squared two proportions power calculation assuming an alpha level of 5%, an intervention group size of 812 with a rate ratio of 0.02 and a control group size of 1,612 with a rate ratio of 0.039 (numbers based on figure 2 from Murad et al.) giving a power of 0.73.”

References

  1. Umpierrez, G.E.; Hellman, R.; Korytkowski, M.T.; Kosiborod, M.; Maynard, G.A.; Montori, V.M.; Seley, J.J.; Van den Berghe, G.; Endocrine, S. Management of hyperglycemia in hospitalized patients in non-critical care setting: an endocrine society clinical practice guideline. J Clin Endocrinol Metab 2012, 97, 16-38, doi:10.1210/jc.2011-2098.
  2. The International Classification of Headache Disorders: 2nd edition. Cephalalgia 2004, 24 Suppl 1, 9-160.
  3. Suh, S.; Park, M.K. Glucocorticoid-Induced Diabetes Mellitus: An Important but Overlooked Problem. Endocrinol Metab (Seoul) 2017, 32, 180-189, doi:10.3803/EnM.2017.32.2.180.
  4. American Diabetes, A. 15. Diabetes Care in the Hospital: Standards of Medical Care in Diabetes-2019. Diabetes Care 2019, 42, S173-S181, doi:10.2337/dc19-S015.
  5. Murad, M.H.; Coburn, J.A.; Coto-Yglesias, F.; Dzyubak, S.; Hazem, A.; Lane, M.A.; Prokop, L.J.; Montori, V.M. Glycemic control in non-critically ill hospitalized patients: a systematic review and meta-analysis. J Clin Endocrinol Metab 2012, 97, 49-58, doi:10.1210/jc.2011-2100.
  6. Blackburn, D.; Hux, J.; Mamdani, M. Quantification of the Risk of Corticosteroid-induced Diabetes Mellitus Among the Elderly. J Gen Intern Med 2002, 17, 717-720.
  7. Gulliford, M.C.; Charlton, J.; Latinovic, R. Risk of diabetes associated with prescribed glucocorticoids in a large population. Diabetes Care 2006, 29, 2728-2729, doi:10.2337/dc06-1499.
  8. Hwang, J.L.; Weiss, R.E. Steroid-induced diabetes: a clinical and molecular approach to understanding and treatment. Diabetes Metab Res Rev 2014, 30, 96-102, doi:10.1002/dmrr.2486.
  9. Liu, X.X.; Zhu, X.M.; Miao, Q.; Ye, H.Y.; Zhang, Z.Y.; Li, Y.M. Hyperglycemia induced by glucocorticoids in nondiabetic patients: a meta-analysis. Annals of nutrition & metabolism 2014, 65, 324-332, doi:10.1159/000365892.
  10. Mealey, B.L. Commentary: managing patients with diabetes: first, do no harm. J Periodontol 2007, 78, 2072-2076, doi:10.1902/jop.2007.070362.
  11. Joy, N.G.; Perkins, J.M.; Mikeladze, M.; Younk, L.; Tate, D.B.; Davis, S.N. Comparative effects of acute hypoglycemia and hyperglycemia on pro-atherothrombotic biomarkers and endothelial function in non-diabetic humans. Journal of diabetes and its complications 2016, 30, 1275-1281, doi:10.1016/j.jdiacomp.2016.06.030.
  12. Stolz, D.; Christ-Crain, M.; Morgenthaler, N.G.; Miedinger, D.; Leuppi, J.; Muller, C.; Bingisser, R.; Struck, J.; Muller, B.; Tamm, M. Plasma pro-adrenomedullin but not plasma pro-endothelin predicts survival in exacerbations of COPD. Chest 2008, 134, 263-272.
  13. Wallace, M.D.; Metzger, N.L. Optimizing the Treatment of Steroid-Induced Hyperglycemia. Ann Pharmacother 2018, 52, 86-90, doi:10.1177/1060028017728297.
  14. Radhakutty, A.; Burt, M.G. MANAGEMENT OF ENDOCRINE DISEASE: Critical review of the evidence underlying management of glucocorticoid-induced hyperglycaemia. Eur J Endocrinol 2018, 10.1530/EJE-18-0315, doi:10.1530/EJE-18-0315.
  15. Schuetz, P.; Hausfater, P.; Amin, D.; Haubitz, S.; Fassler, L.; Grolimund, E.; Kutz, A.; Schild, U.; Caldara, Z.; Regez, K., et al. Optimizing triage and hospitalization in adult general medical emergency patients: the triage project. BMC emergency medicine 2013, 13, 12, doi:10.1186/1471-227X-13-12.
  16. Ceriello, A.; Esposito, K.; Piconi, L.; Ihnat, M.A.; Thorpe, J.E.; Testa, R.; Boemi, M.; Giugliano, D. Oscillating glucose is more deleterious to endothelial function and oxidative stress than mean glucose in normal and type 2 diabetic patients. Diabetes 2008, 57, 1349-1354, doi:10.2337/db08-0063.
  17. Ceriello, A.; Monnier, L.; Owens, D. Glycaemic variability in diabetes: clinical and therapeutic implications. The lancet. Diabetes & endocrinology 2019, 7, 221-230, doi:10.1016/S2213-8587(18)30136-0.

Round 2

Reviewer 1 Report

Authors addressed previous comments and the paper now appears improved.